# Porcine Deltacoronavirus-like Particles Produced by a Single Recombinant Baculovirus Elicit Virus-Specific Immune Responses in Mice

**DOI:** 10.3390/v15051095

**Published:** 2023-04-29

**Authors:** Yangkun Liu, Xueying Han, Yaqi Qiao, Tiejun Wang, Lunguang Yao

**Affiliations:** 1Henan Provincial Engineering and Technology Center of Health Products for Livestock and Poultry, School of Life Science and Agricultural Engineering, Nanyang Normal University, Nanyang 473061, China; liuyangkun@nynu.edu.cn; 2College of Veterinary Medicine, Northwest A&F University, Xianyang 712100, China; 3College of Veterinary Medicine and Engineering, Nanyang Vocational College of Agriculture, Nanyang 473061, China

**Keywords:** porcine deltacoronavirus, virus-like particles, baculovirus, immune response, cytokine, neutralizing antibodies

## Abstract

Porcine deltacoronavirus (PDCoV) causes diarrhea and vomiting in neonatal piglets worldwide and has the potential for cross-species transmission. Therefore, virus-like particles (VLPs) are promising vaccine candidates because of their safety and strong immunogenicity. To the best of our knowledge, the present study reported for the first time the generation of PDCoV VLPs using a baculovirus expression vector system, and electron micrograph analyses revealed that PDCoV VLPs appeared as spherical particles with a diameter similar to that of the native virions. Furthermore, PDCoV VLPs effectively induced mice to produce PDCoV-specific IgG and neutralizing antibodies. In addition, VLPs could stimulate mouse splenocytes to produce high levels of cytokines IL-4 and IFN-γ. Moreover, the combination of PDCoV VLPs and Freund’s adjuvant could improve the level of the immune response. Together, these data showed that PDCoV VLPs could effectively elicit humoral and cellular immunity in mice, laying a solid foundation for developing VLP-based vaccines to prevent PDCoV infections.

## 1. Introduction

Porcine deltacoronavirus (PDCoV), which belongs to the genus *Deltacoronavirus* in the family *Coronaviridae*, is the causative agent of the contagious enteric swine disease [1]. The disease caused by PDCoV is characterized by severe diarrhea, variable vomiting, dehydration, and mortality in neonatal piglets [2]. Since its first report in Hong Kong in 2012, PDCoV has been identified in the United States, Canada, South Korea, mainland China, Thailand, Vietnam, and Japan [3,4,5,6,7]. Hence, PDCoV has recently exhibited a global distribution trend, resulting in substantial economic losses for the global swine industry. Additionally, unlike other porcine enteric coronaviruses, such as porcine epidemic diarrhea virus (PEDV), transmissible gastroenteritis virus (TGEV), and swine acute diarrhea syndrome coronavirus (SADS-CoV), PDCoV can infect multiple species, including pigs, chickens, turkeys, mice, calves, and humans, which may pose a potential threat to human and animal health [8,9,10]. Therefore, it is urgently necessary to develop efficacious preventive methods for preventing and controlling PDCoV.

Vaccine immunization remains the most effective means of disease control; however, no licensed vaccines are currently available for PDCoV. Virus-like particles (VLPs) are assembled from one or more viral structural proteins, and their morphologies and spatial structure are similar to those of the native virions [11]. Because of the advantage of excellent immunogenicity, lack of possible infectivity, and good structural stability, VLPs have been proven to be an effective and reliable platform for vaccine development [12]. To date, several vaccines based on VLPs are commercially available, including human vaccines against human papillomavirus [13], hepatitis B virus [14], and hepatitis E virus [15], as well as veterinary vaccines against porcine circovirus type 2 [16]. These products have greatly improved the development of VLP-based vaccines.

VLP-based vaccines for several coronaviruses, including *Alphacoronavirus* PEDV, *Betacoronaviruses* severe acute respiratory syndrome coronavirus 2 (SARS-CoV-2), and *Gammacoronavirus* avian infectious bronchitis virus (IBV), have been developed and evaluated [17,18,19]. However, there are no reports of VLPs for PDCoV, another member of the Coronaviridae family. Similar to other coronaviruses, the PDCoV genome encodes four major structural proteins: spike (S), envelope (E), membrane (M), and nucleocapsid (N) proteins [20]. According to studies on other coronaviruses, coronavirus VLPs generally consist of the proteins of M + E or M + E + S [17,18,19,21]. Thus, we speculate that co-expression of S, M, and E is sufficient for the formation of PDCoV VLPs. This study investigated the production and characterization of PDCoV VLPs from recombinant insect cells co-expressing PDCoV proteins (M, S, and E). The immunogenicity of VLPs was also evaluated in mice, which suggests that the VLPs obtained in this study could be used for vaccine development against PDCoV infection.

## 2. Materials and Methods

### 2.1. Bacterial Strains, Plasmids, Cells, and Viruses

*E. coli* SW106 AcMultiBac, which contains AcBacmid, pHelper, and pGB2Ωinv, was constructed in a previous study [22]. pFBDM plasmid and the AcMultiBac system that uses gentamicin resistance selection following Tn7 transposition were maintained in our laboratory.

*Spodoptera frugiperda* 9 (Sf9) cells were cultured in Sf-900 III serum-free medium (SFM; Thermo Fisher Scientific, Waltham, MA, USA) at 27 °C. Swine testis (ST) cells were maintained in Dulbecco’s modified Eagle’s medium (DMEM; Invitrogen, Carlsbad, CA, USA), supplemented with 10% fetal bovine serum (FBS; Biological Industries, Kibbutz, Israel) at 37 °C with 5% CO_2_. The PDCoV strain of HeN (GenBank accession number MN942260.1) was kindly provided by Professor Enqi Du (Northwest A&F University, Shanxi, China). Mouse anti-PDCoV polyclonal antibodies were prepared and stored in our laboratory.

### 2.2. Construction of the MSE Triple Expression Plasmid

To obtain a new triple expression plasmid, the p10 promoter and HSV TK polyadenylation sequence were digested from the pFBDM plasmid by *Pme*I and *Spe*I restriction enzymes and subcloned into another pFBDM plasmid at the BstZ17I and SpeI sites to generate the pFBTM recombinant plasmid. First, the full-length S, M, and E genes from PDCoV isolate with C-terminal 6 × His tag (for S), HA tag (for M), or Flag tag (for E) were codon-optimized and synthesized (Genscript, Nanjing, China) for high-level expression in Sf9 cells. Then, the codon-optimized S, M, and E genes were cloned into a single pFBTM plasmid, each gene within its expression cassette. The S gene was inserted into the *Sma*I and *Kpn*I restriction sites under the control of the p10 promoter, the M gene was cloned into the *Bam*HI and *Sal*I restriction sites under the control of the polyhedron (polh) promoter, and the E gene was inserted into the *Xho*I and *Nhe*I restriction sites under the control of the p10 promoter. Finally, DNA sequencing generated and verified the triple expression plasmid named pFBTM-MSE (Tsingke, Beijing, China).

### 2.3. Generation of Recombinant Baculovirus

The recombinant plasmid pFBTM-MSE was transformed into *E. coli* SW106 AcMultiBac for transposition into the genome of AcMultiBac, followed by the antibiotic selection and blue-white selection. Afterward, the white *E. coli* colonies containing the recombinant bacmid were identified through polymerase chain reaction (PCR) technology using primer pairs MF/MR, SF/SR, and EF/ER (Table 1). The resulting recombinant strain was designated as rBacmid-MSE.

Recombinant baculovirus was generated, as previously described [22]. Briefly, the recombinant bacterial strains containing rBacmid-MSE were cultured at 30 °C, shaken at 180 rpm for 12 h, and collected by centrifugation. The bacterial pellet was washed with distilled ultrapure water three times. Then, the pellet was resuspended in 1 mL of Sf-900 III SFM and adjusted to different densities (10^5^–10^8^ cells/mL). Sf9 cells were incubated overnight in a 24-well plate (70–80% confluent single layer). After removing the medium, 500 μL of bacterial cells were added to the corresponding wells at different concentrations. After culturing at 27 °C for 4–5 h, the bacteria in each cell well were removed by washing with Sf-900 III SFM. Then, 500 μL fresh Sf-900 III SFM was added into each well and incubated for 3–5 days. Cytopathic effect (CPE) was observed using a microscope (Zeiss Axioskop-40, Oberkochen, Germany) to determine the production of recombinant baculovirus. The supernatant of cells transfected with the recombinant strain was harvested and passaged in Sf9 cells; the resulting recombinant baculovirus was named rBV-MSE. The baculovirus stock titer was determined according to the instructions of the Baculovirus Rapid Titer Kit (TaKaRa, Tokyo, Japan).

### 2.4. Indirect Immunofluorescence Assay

To detect viral protein expression by the recombinant baculovirus, Sf9 cells were grown in a 24-well plate and infected with rBV-MSE at a multiplicity of infection (MOI) of 2. At 48 h post-infection (hpi), the cells were processed for indirect immunofluorescence assay (IFA). Briefly, Sf9 cells were washed and fixed with 4% paraformaldehyde for 10 min at room temperature. After blocking with 5% BSA at 37 °C for 2 h, the cells were then incubated with mouse anti-His antibodies (Boster, Wuhan, China), mouse anti-HA antibodies (Boster, Wuhan, China), or mouse anti-Flag antibodies (Boster, Wuhan, China) for 2 h at 37 °C. After washing with phosphate buffer saline (PBS), Alexa Fluor 594 conjugated goat anti-mouse IgG (H + L) (Boster, Wuhan, China) was added to the cells and incubated at 37 °C for 1 h. Finally, cells were observed under a fluorescence microscope (Zeiss Axioskop-40, Oberkochen, Germany).

### 2.5. Production and Purification of VLPs

Sf9 suspension cells (2 × 10^6^ cells/mL) were cultured in 500 mL polycarbonate Erlenmeyer flasks and incubated in an orbital shaker incubator at 125 rpm and 27 °C. The cultures were infected with rBV-MSE at an MOI of 5. At 72 hpi, the cells were harvested by centrifugation (3000× *g*, 10 min) and lysed by sonication. The lysed Sf9 cells were centrifuged at 12,000× *g* for 10 min. Then, the supernatant was centrifuged at 35,000 rpm in an SW41 Ti rotor (Beckman, Brea, CA, USA) for 2 h at 4 °C. The collected precipitates were dissolved in PBS and the solution was loaded on the top surface of 20–40–60% (*w*/*v*) sucrose gradient and then centrifuged at 35,000 rpm for 4 h at 4 °C. The white band between the 40% and 60% sucrose liquid levels was collected and dissolved in PBS. Finally, the solution concentration was measured using the BCA protein assay kit (Sangon, Shanghai, China) and subjected to Western blotting analysis or electron microscopy.

### 2.6. Western Blotting Analysis

Purified VLPs or cell lysates were separated using 12% SDS-PAGE gels and then transferred onto a polyvinylidene fluoride (PVDF) membrane (Millipore Corp, Billerica, MA, USA). The resulting membrane was blocked with 5% skim milk in tris buffered saline with tween 20 (TBST) overnight at 4 °C. Subsequently, the membrane was incubated with a 1:3000 dilution of mouse polyclonal antibodies against PDCoV or mouse anti-His antibodies, mouse anti-HA antibodies, and mouse anti-Flag antibodies for 2 h at room temperature. Unbound antibodies were removed by three washes of 5 min each in TBST buffer. Next, the membrane was treated with horseradish peroxidase (HRP)-conjugated goat anti-mouse IgG (H + L) (Boster, Wuhan, China) for 1 h at room temperature. Finally, images were captured using a luminescent imaging system (Amersham ImageQuant 800; Cytiva, Uppsala, Sweden).

### 2.7. Transmission Electron Microscopy

The purified VLPs samples were adsorbed onto a carbon-coated grid for 2 min and the residual liquid was removed by blotting with filter paper. Next, the grid was negatively stained with 2% phosphotungstic acid (pH 6.45) for 1 min, air-dried, and examined with a Hitachi H-7600 transmission electron microscope (TEM) at 80 kV. Then, the average size of particles was analyzed by dynamic light scattering (DLS) with a Zetasizer Nano ZS instrument (Malvern Instruments Ltd., Malvern, UK). Three repeats were performed.

### 2.8. Immunization of Mice

A total of 20 8-week-old female BALB/C mice (Autobio, Zhengzhou, China) were randomly divided into 4 groups (*n* = 5 per group). The mice from the PBS and Freund groups were inoculated with PBS and an emulsion of Freund’s adjuvant plus PBS at the volume ratio of 1:1. The mice from the VLPs group were inoculated with the solution of VLPs (10 μg VLPs/mouse) and the mice from the VLPs/Freund group were immunized with the emulsion of Freund’s adjuvant plus VLPs solution (10 μg VLPs/mouse) at the volume ratio of 1:1. These mice were inoculated intramuscularly with a volume of 100 μL in the quadriceps of legs. The mice were immunized twice at an interval of 2 weeks. Freund’s complete and incomplete adjuvants were used for the first and second immunization, respectively. Blood was collected through the tail vein weekly, and spleens were collected at week 5 to isolate lymphocytes. All animal experiments were conducted following the regulations of the Animal Research Ethics Board of Nanyang Normal University (No. NYNU-2022-016).

### 2.9. Determination of IgG Antibodies

ELISA was used to determine antibody titers of PDCoV-specific IgG in serum from vaccinated mice. Briefly, Costar polystyrene high binding 96-well plates (Corning, New York, NY, USA) were coated with 100 µL of PDCoV (10^5^ TCID_50_/mL) overnight at 4 °C and blocked with 5% skim milk for 1 h at 37 °C. Serum samples were 1:20 diluted and added to the coated plates (100 μL/well). Plates were incubated at 37 °C for 1 h, followed by incubation with HRP-conjugated goat anti-mouse IgG at 37 °C for 1 h. Reactions were developed with 3,3′,5,5′-tetramethylbenzidine (TMB) for 15 min at 25 °C and terminated with 2M H_2_SO_4_. The microplate analyzer detected OD_450 nm_ absorbance value within 15 min. The cutoff value was determined by counting the mean OD value of serum from the mice before immunization plus three standard deviations (SD).

### 2.10. Detection of Neutralizing Antibodies

Two weeks after the booster immunization, a virus neutralization test (VNT) was performed using PDCoV HeN to determine the levels of neutralizing antibodies (NAbs) in the serum of vaccinated mice. Briefly, serum was heated at 56 °C for 30 min for complement inactivation. Next, 100 µL of twofold serially diluted serum was incubated with an equal volume of DMEM containing 100 TCID_50_ PDCoV virus at 37 °C for 1 h. Then, a 200 µL mixture was added to the ST cells in eight 96-well cell culture plate wells. CPE was observed for 5–7 days, and the NAb titer was determined according to the Reed-Muench method.

### 2.11. Cytokine Release Assay

Three weeks after the booster immunization, the spleens of mice were obtained aseptically and grounded in a 35 mm Petri dish containing 5 mL of pig 1× lymphocyte separation medium (DAKEWE, Shenzhen, China); suspensions were filtered using 40 µm filters into a new tube and were centrifuged at 600× *g* for 30 min. The splenocytes were collected and washed with RPMI 1640 medium (Pricella, Wuhan, China) and then added to 24-well plates with 1 × 10^6^ cells per well and then treated with PDCoV VLPs (5 μg/mL) at 37 °C for 72 h. According to the manufacturer’s instructions, the IFN-γ and IL-4 in culture supernatants were evaluated with ELISA kits (Neobioscience, Shenzhen, China).

### 2.12. Statistical Analysis

Statistical analyses were performed using GraphPad Prism 5 software (GraphPad Software Inc., San Diego, CA, USA), and one-way analysis of variance (ANOVA) was used to determine the differences. Data are expressed as the mean ± SD. Statistical significance was designated for differences with *p*-values < 0.05 (* *p* < 0.05; ** *p* < 0.01).

## 3. Results

### 3.1. Generation and Identification of Recombinant Bacmids

To express the PDCoV M, S, and E proteins in Sf9 cells, a new triple expression plasmid pFBTM containing two pP10 promoters, one pPH promoter, and three multiple cloning sites was constructed. The M, S, and E genes were then cloned into the pFBTM vector to generate the recombinant plasmid pFBTM-MSE (Figure 1). After verification by Sanger sequencing, the pFBTM-MSE plasmids were transformed into *E. coli* SW106 AcMultiBac competent cells, and the recombinant bacmids were extracted and identified through PCR. The results showed that the amplification products containing the PDCoV M, S, and E fragments were approximately 651 bp, 3477 bp, and 249 bp, respectively, consistent with the expected sizes (Appendix A). Therefore, the recombinant bacmids co-expressing M, S, and E were successfully constructed.

### 3.2. Expression and Analysis of the Recombinant Protein in Sf9 Insect Cells

To obtain the recombinant baculovirus, Sf9 cells were infected with the recombinant bacterial strains containing rBacmid-MSE. The pathological changes in Sf9 cells were observed at 96 hpi. As shown in Figure 2A, compared with normal cells, Sf9 cells infected with rBV-MSE display an obvious CPE, including an increase in cell and nucleus diameter, the appearance of cell debris, and detachment from the plate. The titers of the second-generation rBV-MSE were 2.1 × 10^7^ PFU/mL. Sf9 suspension cells were then infected with rBV-MSE at an MOI of 5 and harvested at 24, 48, 72, 96, and 120 hpi. The expression of target proteins was then detected with Western blotting analysis using mouse polyclonal antibodies against PDCoV. As shown in Figure 2B, the bands of S (180 kDa), E (12 kDa), and M (23 kDa) recombinant proteins were determined in the lysate of infected Sf9 cells but not normal Sf9 cells. Furthermore, the target proteins were detected in cell lysates harvested at 48 to 120 hpi, with the highest expression of the protein at 72 hpi. Regarding further evidence, the expression of the S, M, and E proteins was confirmed in Sf9 cells infected with rBV-MSE using IFA (Figure 2C), whereas there was no specific fluorescence in normal cells (Appendix A). These data demonstrated that the S, M, and E proteins were successfully expressed in Sf9 cells.

### 3.3. Purification and Characterization of PDCoV VLPs

To obtain VLPs with high purity, Sf9 cells were infected with recombinant baculoviruses, and the proteins were purified by sucrose density gradient ultracentrifugation. TEM analysis showed the VLPs were in enveloped spherical shape (Figure 3A). Similarly, the purified VLPs were found to be homogenous by DLS, and the majority VLPs were approximately 100–120 nm in diameter (Figure 3B). To further determine whether the VLPs were successfully assembled, the components of VLPs were identified through Western blotting analysis. Figure 3B shows that three target bands corresponding to the expected size were observed simultaneously. These results indicate that PDCoV VLPs autonomously assemble in insect cells infected with recombinant baculoviruses, and they are structurally similar to the native virions [23].

### 3.4. Determination of PDCoV-Specific IgG and NAbs in Mice

To determine the immunogenicity of PDCoV VLPs, mice were immunized according to the protocol shown in Figure 4A. ELISA was used to measure the PDCoV-specific IgG antibody levels in the serum of vaccinated mice. Two weeks after the first immunization, PDCoV-specific IgG antibodies in the mice inoculated with VLPs and VLPs/Freund were detected as positive, and then the antibody level gradually increased. No PDCoV-specific antibodies were detected in PBS and Freund groups during the experiment (Figure 4B). Analysis of PDCoV NAbs in the serum of vaccinated mice showed that, although both VLPs and VLPs/Freund induced PDCoV NAb production, VLPs/Freund elicited significantly increased PDCoV NAb levels compared with VLPs (*p* < 0.01; Figure 4C).

### 3.5. Analysis of Cytokine Production in Splenocytes

To investigate cellular immune responses, we examined Th1-type (IFN-γ) and Th2-type (IL-4) cytokines in the splenocyte supernatants of vaccinated mice. As shown in Figure 5A, the concentration of IFN-γ in VLPs and VLPs/Freund groups was significantly increased compared with that of the PBS and Freund groups (*p* < 0.01). Furthermore, the level of IFN-γ in the VLPs/Freund group was higher than that of the VLPs group (*p* < 0.05). Additionally, the level of IL-4 was significantly higher in the VLPs/Freund group than in the VLP, PBS, and Freund groups (*p* < 0.01). Therefore, PDCoV VLPs promoted IFN-γ and IL-4 production, inducing an antigen-specific cellular immune response in mice.

## 4. Discussion

The rapid emergence and widespread transmission of PDCoV poses serious health threats to humans and animals worldwide, leading to an urgent need for effective vaccine development approaches. Since PDCoV is a recently emerged viral pathogen, no commercial vaccines are available. To date, several strategies for PDCoV vaccine development, including inactivated vaccines [24], live-attenuated vaccines [25], and viral vector vaccines [26], have been evaluated. However, each vaccine has disadvantages, limiting its application and effectiveness in controlling PDCoV. VLPs have demonstrated enhanced immune responses and greater protection than traditional vaccines. Still, they are not infectious or replicative due to the lack of the viral genome, making them an ideal vaccine candidate against various viruses [11,12]. Additionally, as VLP is versatile to genetic and chemical modification and self-adjuvants, VLP-based vaccines could easily become one of the most effective vaccines against coronaviruses. Therefore, it should be developed with greater effort.

Previous studies have reported the production of CoV-like particles using the baculovirus-insect cell expression system (BEVS) [27]. PEDV, IBV, and SARS-CoV-2 VLPs were successfully prepared by co-expression of S, E, and M proteins in Sf9 insect cells, with shapes and sizes similar to those of the naive virus. Compared with other expression systems (mainly bacterial, yeast, or mammalian expression systems), BEVS has many advantages in assembling VLPs, including inherent safety, the capacity to accommodate large foreign genes, proper protein folding, and post-translational processing [28]. Therefore, BEVS could serve as a platform for producing PDCoV VLPs. In this study, the recombinant baculoviruses containing the M, S, and E genes of PDCoV were constructed and used to infect Sf9 insect cells. The results of IFA and Western blotting assays showed that the M, S, and E proteins were co-expressed in Sf9 cells infected with rBV-MSE. Subsequently, PDCoV VLPs were efficiently purified using sucrose gradient purification. The morphology of the purified VLPs was similar with the native virions under TEM. These results indicated that the BEVS is suitable for producing PDCoV VLPs.

VLP-based vaccines were reported to interact with both innate and adaptive immune cells, thus inducing strong humoral cellular immune responses [29,30,31]. To evaluate the immune responses induced by PDCoV VLPs, we immunized mice with VLP vaccines formulated by mixing PDCoV VLPs (10 μg) with or without Freund’s adjuvant. Humoral immunity plays an important role in the fight against coronavirus infection [32]. The ELISA data showed that VLPs could induce an antigen-specific IgG response and cause mice to produce neutralizing antibodies. VLPs plus Freund’s adjuvant elicited a stronger IgG response and neutralizing antibodies, which suggested that the adjuvant could modulate the immune response induced by VLPs. In addition, mice inoculated with VLPs or adjuvanted VLPs produced high levels of IFN-γ (Th1-type cytokines) and IL-4 (Th2-type cytokines), which indicated that VLPs triggered Th1/Th2-mediated cellular immune response. These results indicated that PDCoV VLPs could effectively stimulate the cellular and humoral response in mice, which exhibit a good immunogenicity.

To the best of our knowledge, this study reports for the first time the production of PDCoV VLPs using a single recombinant baculovirus; however, there are some limitations in this study. For example, the conditions for preparing PDCoV VLPs, including baculovirus infection time, MOI, modifications of the signal peptides, and so on, should be further optimized. In addition, studies of the immunogenicity and protective effect against PDCoV in pigs are lacking due to the limited experiment conditions. It is generally accepted that passive immunity transferred from the sows to the piglets through the colostrum and milk is critical for protecting piglets from enterovirus infection (e.g., PEDV and TGEV) [33,34]. Passive immunity is mainly achieved through high titers of IgG antibodies in colostrum within the first 24–48 h after birth, and then persistent supply of secretory IgA (sIgA) antibodies in milk throughout lactation. Previous studies suggest that maternal sIgA, IgG, and VN antibodies contribute to the protection of the neonatal pig against PDCoV infections [24]. Therefore, the proper vaccine strategies in pregnant swine to induce both systemic antibody and maternal secretory IgA in milk should be taken into account.

In conclusion, we reported here that co-expressing PDCoV M, S, and E protein in insect cells were assembled into PDCoV VLPs, which could efficiently induce PDCoV-specific humoral immune responses and cellular immune response in mice. Therefore, the PDCoV VLPs generated in this study have greater potential for vaccine development to control PDCoV.

## Figures and Tables

**Figure 1 viruses-15-01095-f001:**
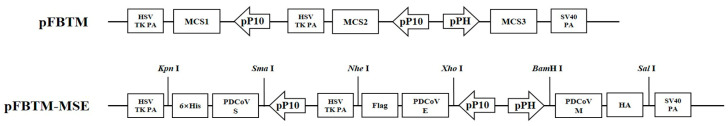
Schematic representation of the MSE triple expression plasmids. pPH, polyhedrin promoter of baculovirus; pP10, p10 promoter of baculovirus; 6 × His, hexahistidine (HHHHHH) peptide tag; Flag, DYKDDDDK peptide tag; and HA, YPYDVPDYA peptide tag.

**Figure 2 viruses-15-01095-f002:**
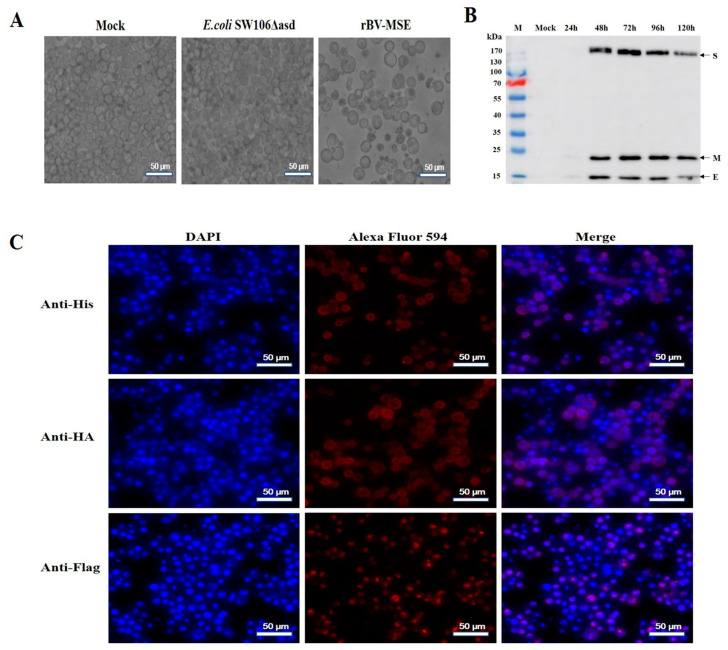
Expression and analysis of the recombinant protein in Sf9 cells. (**A**) Cytopathic effect of Sf9 cells after transfection with rBacmid-MSE for 96 h. Left, untransfected Sf9 cells. Middle, Sf9 cells transfected with *E. coli* SW106Δasd. Right, Sf9 cells transfected with rBacmid-MSE. (**B**) Western blotting analysis of the recombinant protein expression in Sf9 cells. Cells were harvested at 24, 48, 72, 96, and 120 hpi. Protein expression was detected with mouse polyclonal antibodies against PDCoV and HRP-labeled goat anti-mouse IgG (H + L). Mock, normal Sf9 cells. (**C**) Identification of the recombinant proteins expressed in Sf9 cells using IFA. Sf9 cells infected with rBV-MSE were subjected to immunostaining using His, HA, and Flag antibodies. These cells were detected using fluorescence microscopy.

**Figure 3 viruses-15-01095-f003:**
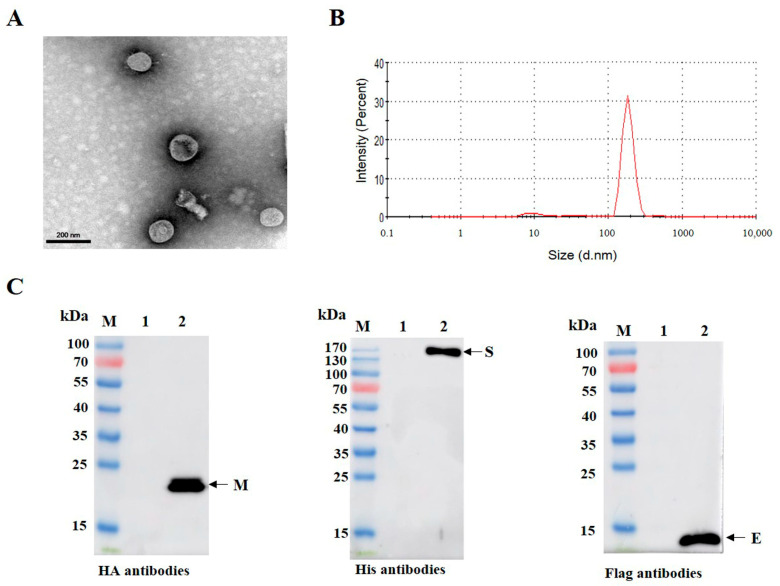
Identification of PDCoV VLPs using TEM and Western blotting analysis. (**A**) Electron microscopic images of purified PDCoV VLPs. H-7600 was used to observe the morphology and size of PDCoV VLPs. (**B**) Determination of size distribution of purified VLPs with dynamic light scattering. (**C**) Using His, HA, and Flag antibodies, sucrose-purified VLPs were probed with Western blotting. M, 10–170 kDa protein marker; 1, PBS control; 2, sucrose-purified VLPs.

**Figure 4 viruses-15-01095-f004:**
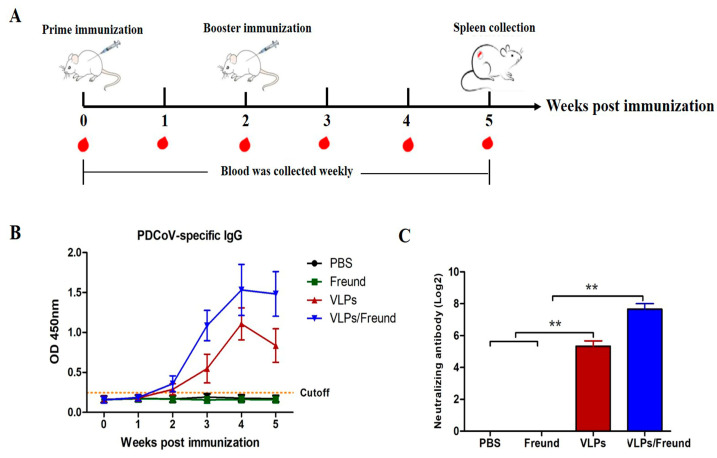
Determination of PDCoV-specific IgG and NAb in mice. (**A**) Mouse immunization protocol. (**B**) PDCoV-specific IgG antibodies. The antibody level was detected using the indirect ELISA method. OD_450 nm_ greater than or equal to 0.2256 was positive; OD_450 nm_ less than 0.2256 was negative. (**C**) Detection of neutralizing antibodies. The results represent the mean value (Log_2_ value) ± SD (*n* = 3), and differences were considered to be significant at ** *p* < 0.01.

**Figure 5 viruses-15-01095-f005:**
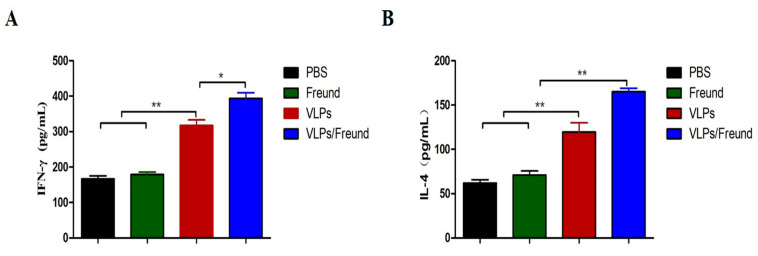
Analysis of cytokine production in splenocytes. IFN-γ (**A**) and IL-4 (**B**) levels from splenocyte supernatants were detected using ELISA. Statistical differences between groups are shown as * *p* < 0.05 or ** *p* < 0.01. The data are expressed as the mean ± SD (*n* = 3).

**Table 1 viruses-15-01095-t001:** Primer sequences used for PCR analysis.

Primers	Sequence (5′-3′)	PCR Product Size
MF	ATGTCCGATGCTGAGGAGTGG	651 bp
MR	CATGTACTTATACAGTCGAG	
SF	ATGCAACGAGCTTTGTTAAT	3477 bp
SR	CCATTCTTTGAACTTAAAGGAC	
EF	ATGGTAGTCGACGACTGGGCC	249 bp
ER	CACGTAATGCGTGTTCCTTG	

## Data Availability

Not applicable.

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
