# Peer review of "Porcine Deltacoronavirus-like Particles Produced by a Single Recombinant Baculovirus Elicit Virus-Specific Immune Responses in Mice"

_viruses, 2023, doi:10.3390/v15051095_

Round 1

Reviewer 1 Report

Your work won't work in a pig.

While increased serum IgG may translate into increased maternal antibody available to the suckling pig, its protection lasts only a couple of days until secretory IgA appears in the milk secretions.  IgG in sow's milk is largely digested in piglets' stomach.  sIgA is protected from stomach digestion by the secretory piece that joins two IgA monomers.  Further, sIgA is produced locally in the mammary gland, not systemically.  Thus, increases in serum antibody are not a measure of piglet protection.  Please review the literature about sow secretory antibody and include it in your discussion.  Otherwise, I will recommend rejection of the manuscript.

Author Response

Your work won't work in a pig.

While increased serum IgG may translate into increased maternal antibody available to the suckling pig, its protection lasts only a couple of days until secretory IgA appears in the milk secretions.  IgG in sow's milk is largely digested in piglets' stomach.  sIgA is protected from stomach digestion by the secretory piece that joins two IgA monomers.  Further, sIgA is produced locally in the mammary gland, not systemically.  Thus, increases in serum antibody are not a measure of piglet protection.  Please review the literature about sow secretory antibody and include it in your discussion.  Otherwise, I will recommend rejection of the manuscript.

Response: Thanks for your kind suggestion. We have added the literature about sow secretory antibody in our discussion.

Reviewer 2 Report

Yangkun and cols produced and evaluated virus-like particles (VLPs) of porcine deltaconavirus (PDCoV). The authors used baculovirus expression to produce VLPs of PDCoV expressing three viral proteins. The authors also used electron microscopy analysis to characterize the morphological characteristics of VLPs. Finally, the VLPs were used to immunize mice. The VLPs were successfully produced and could induce neutralizing antibodies and cytokine production. The interesting and clear study provides new information for generating further vaccines to control PDCoV.

I found several grammatical mistakes, especially in the discussion section. Please check and correct accordingly. 

The authors have shown the expression of S, E, and M proteins in the VLPs, but confirming that mice produce antibodies against all proteins will provide additional information about the immunogenicity of all proteins contained in the VLPs. Neutralizing antibodies are mainly directed to the S protein. In the same line, it is necessary to demonstrate the participation of each protein in cytokine production. The estimation with the VLPs is important but insufficient, considering that VLPs contain three proteins (S, M, and E).

The discussion must be modified. The results of this study do not support that VLPs induce strong cellular and humoral responses (lines 331-332). Additional controls are necessary to support the statement. 

It would be great if the authors tested the VLPs in pigs. Most of the time response in mice could be different in pigs. 

Author Response

Yangkun and cols produced and evaluated virus-like particles (VLPs) of porcine deltaconavirus (PDCoV). The authors used baculovirus expression to produce VLPs of PDCoV expressing three viral proteins. The authors also used electron microscopy analysis to characterize the morphological characteristics of VLPs. Finally, the VLPs were used to immunize mice. The VLPs were successfully produced and could induce neutralizing antibodies and cytokine production. The interesting and clear study provides new information for generating further vaccines to control PDCoV.

I found several grammatical mistakes, especially in the discussion section. Please check and correct accordingly. 

Response: Thanks for your kind suggestion. We checked the manuscript carefully, and then got LetPub (www.letpub.com) for its linguistic assistance to check the grammatical mistakes in this manuscript.

The authors have shown the expression of S, E, and M proteins in the VLPs, but confirming that mice produce antibodies against all proteins will provide additional information about the immunogenicity of all proteins contained in the VLPs. Neutralizing antibodies are mainly directed to the S protein. In the same line, it is necessary to demonstrate the participation of each protein in cytokine production. The estimation with the VLPs is important but insufficient, considering that VLPs contain three proteins (S, M, and E).

Response: Thanks for your kind suggestion. We fully agree with your opinion that it is necessary to demonstrate the participation of each protein in antibodies and cytokine production. However, the aim of this study is to determine whether co-expression of S, M, and E could produce PDCoV VLPs, and whether VLP could induce virus-specific immune responses in mice, thus we think exploring the function of each protein is another systematic and independent study need to be done.   

The discussion must be modified. The results of this study do not support that VLPs induce strong cellular and humoral responses (lines 331-332). Additional controls are necessary to support the statement.

Response: Thanks for your kind suggestion. We have revised the discussion section in the revised manuscript text.

It would be great if the authors tested the VLPs in pigs. Most of the time response in mice could be different in pigs. 

Response: Thanks for your kind suggestion. We fully agree with your opinion, studies of the immunogenicity and protective efficacy of the VLPs against PDCoV in pigs are ongoing.

Reviewer 3 Report

This manuscript from Liu et al details the production and testing of a novel VLP vaccine against Porcine deltacoronavirus (PDCoV), a newly emergent pathogen with demonstrable impacts on the worldwide swine industry and the potential for further spread due to its ability to infect multiple species.  The group used the AcMultiBac EColi system to generate recombinant baculovirus as a delivery vector to express PDCoV M, E and S proteins as well as the assemble components to generate VLP formation.  Upon verification of VLP formation and PDCoV protein expression, the vaccine generated significant adaptive immune responses (virus-specific and neutralizing antibodies and activated splenocytes) in BALB/c mice with a prime/ boost regimen, and its effects were accentuated by the inclusion of Freund's adjuvant. 

The paper as a whole was well written and the experiments were clearly explained, and I only have minor comments to be addressed prior to acceptance for publication:

1)  On page 4, the authors should specify the specific microtiter plates used for ELISA experiments and their source.

2) On page 5, the authors should further detail their methods used to obtain a single-cell splenocyte suspension (including any purification steps).

3) In the methods or in section 3.1, the authors should specify the components of rBacmid that allow for VLP formation.

4) In figure 2A, the authors should include an image of Sf9 cells infected with E Coli that does not contain baculovirus to demonstrate any cytopathic effects of co-culture of Sf9 cells with E Coli.

5) In Figure 2B, the authors should specify what their NC (negative control) is.

6) In figure 2C, the authors should include images of uninfected Sf9 cells to ensure specific fluorescent signal.

7) In figure 3A, the authors should quantify VLP particle formation (such as by size or by ratio of complete to incomplete particles) if possible.

8) In figure 3B, the authors should include negative controls in their western blots to ensure specific signal of the tag antibodies.

9) In the discussion, the authors should briefly outline further steps needed before beginning testing of this vaccine in pigs.

Author Response

This manuscript from Liu et al details the production and testing of a novel VLP vaccine against Porcine deltacoronavirus (PDCoV), a newly emergent pathogen with demonstrable impacts on the worldwide swine industry and the potential for further spread due to its ability to infect multiple species.  The group used the AcMultiBac EColi system to generate recombinant baculovirus as a delivery vector to express PDCoV M, E and S proteins as well as the assemble components to generate VLP formation.  Upon verification of VLP formation and PDCoV protein expression, the vaccine generated significant adaptive immune responses (virus-specific and neutralizing antibodies and activated splenocytes) in BALB/c mice with a prime/ boost regimen, and its effects were accentuated by the inclusion of Freund's adjuvant. 

The paper as a whole was well written and the experiments were clearly explained, and I only have minor comments to be addressed prior to acceptance for publication:

1)  On page 4, the authors should specify the specific microtiter plates used for ELISA experiments and their source.

Response: Thanks for your kind suggestion. We have added the details of ELISA plates in the revised manuscript text in page 4.

2) On page 5, the authors should further detail their methods used to obtain a single-cell splenocyte suspension (including any purification steps).

Response: Thanks for your kind suggestion. We have added the details of our methods used to obtain a single-cell splenocyte suspension in the revised manuscript text in page 5.

3) In the methods or in section 3.1, the authors should specify the components of rBacmid that allow for VLP formation.

Response: Thanks for your kind suggestion. We have added the components of rBacmid in the revised manuscript text in page 5.

4) In figure 2A, the authors should include an image of Sf9 cells infected with E Coli that does not contain baculovirus to demonstrate any cytopathic effects of co-culture of Sf9 cells with E Coli.

Response: Thanks for your kind suggestion. We have added an image of Sf9 cells transfected with E. coli that does not contain baculovirus (E. coli SW106Δasd) in Figure 2A.

5) In Figure 2B, the authors should specify what their NC (negative control) is.

Response: Thanks for your kind suggestion. The NC in Figure 2B represent Normal Sf9 cells (Mock), we have added it in the revised manuscript text. 

6) In figure 2C, the authors should include images of uninfected Sf9 cells to ensure specific fluorescent signal.

Response: Thanks for your kind suggestion. We have added the images of uninfected Sf9 cells in Supplementary Figure S2.

7) In figure 3A, the authors should quantify VLP particle formation (such as by size or by ratio of complete to incomplete particles) if possible.

Response: Thanks for your kind suggestion. The average size of VLP particles were analyzed by dynamic light scattering (DLS) with a Zetasizer Nano ZS instrument (Malvern Instruments Ltd., Malvern, UK). The results were shown in Figure 3B in the revised manuscript text.

8) In figure 3B, the authors should include negative controls in their western blots to ensure specific signal of the tag antibodies.

Response: Thanks for your kind suggestion. We have added the negative controls in our western blots in the revised manuscript text.

9) In the discussion, the authors should briefly outline further steps needed before beginning testing of this vaccine in pigs.

Response: Thanks for your kind suggestion. We have briefly stated further steps needed before beginning testing of this vaccine in pigs in the revised manuscript text.

Round 2

Reviewer 1 Report

First, please know the research content of this manuscript is interesting. Its value would be greatly enhanced if the authors would explain its shortcomings and teach their readers about the origins of maternally-derived antibody in swine. 

Second, the authors say that they "have added the literature about sow secretory antibody in our discussion." I disagree. Rather, they made a word salad.   The discussion states, "Previous studies showed that secretory IgA antibodies transferred from the sows to the piglets have been considered as the primary mechanism to achieve an adequate immune protection against swine enterovirus infection [33]. Considering that the production of IgA in the colostrum and milk was associated with enteric cavity, it is also necessary to optimize the inoculation routes."

A word search of "swine enterovirus" in the paper (Langel, et al., 2020) found no matches.  Thus, it is misleading to cite this reference.

"Optimizing ... vaccine efficacy" is found in the last sentence of the Langel, et al., 2020).

These lead me to conclude that the authors are not the least bit familiar with porcine maternal transfer of antibody.  Thus, they will be hard-pressed to explain the value of their research in the context of swine immunity.

Author Response

First, please know the research content of this manuscript is interesting. Its value would be greatly enhanced if the authors would explain its shortcomings and teach their readers about the origins of maternally-derived antibody in swine. 

Response:Thank you for your kind suggestion. We have added the limitations and the passive immunity in the revised manuscript.  

Second, the authors say that they "have added the literature about sow secretory antibody in our discussion." I disagree. Rather, they made a word salad. The discussion states, "Previous studies showed that secretory IgA antibodies transferred from the sows to the piglets have been considered as the primary mechanism to achieve an adequate immune protection against swine enterovirus infection [33]. Considering that the production of IgA in the colostrum and milk was associated with enteric cavity, it is also necessary to optimize the inoculation routes."

A word search of "swine enterovirus" in the paper (Langel, et al., 2020) found no matches.  Thus, it is misleading to cite this reference.

"Optimizing ... vaccine efficacy" is found in the last sentence of the Langel, et al., 2020).

These lead me to conclude that the authors are not the least bit familiar with porcine maternal transfer of antibody. Thus, they will be hard-pressed to explain the value of their research in the context of swine immunity.

Response:Thank you for your kind suggestion. According to the statement that “sIgA antibodies play a major role in conferring passive lactogenic protection against enteric pathogens in suckling neonates” in the discussion of the Langel, et al., 2020, we write the word "swine enterovirus" in our paper, so we are very sorry for our incorrect writing has confused you. We have re-written this section in the revised manuscript.

Reviewer 2 Report

I have not additional comments.

Author Response

Comments: I have not additional comments.

Response: Special thanks to you for your good comments.